# Icings of the Kunlun Mountains on the Northern Margin of the Qinghai-Tibet Plateau, Western China: Origins, Hydrology and Distribution

Leonid Gagarin [1,2], Qingbai Wu [1,*], Wei Cao [1] and Guanli Jiang [1]

1    State Key Laboratory of Frozen Soils Engineering, Northwest Institute of Eco-Environment and Resources, Chinese Academy of Sciences, 320 W. Donggang Rd., Lanzhou 730000, China; gagarinleo@yandex.ru (L.G.); caowei@lzb.ac.cn (W.C.); atos@lzb.ac.cn (G.J.)

2    Melnikov Permafrost Institute, Russian Academy of Sciences (RAS), 36 Merzlotnaya St., 677010 Yakutsk, Russia

*    Correspondence: qbwu@lzb.ac.cn; Tel.: +86-931-4967284

**Abstract:** Icing/Aufeis processes are a typical feature of permafrost hydrology in mountainous regions. Regional databases of Aufeis have been compiled since the 2010. In this study, we attempted to create an initial Aufeis database for the Qinghai-Tibet Plateau (QTP) to evaluate the patterns of the icing processes in the arid and high mountain regions at low latitudes. In this article, the icings/Aufeis in the Kunlun Mountains on the northern edge of the QTP were investigated. A total of 65 Landsat 8 Operational Land Imager images for 2017–2020 of the key sites were acquired. Icings occur at elevations of 2500–5400 m a. s. l. More than 1600 Aufeis were identified with a total ice-surface area of 2670 km$^2$. About 88% of these areas are related to a gigantic Aufeis (tarin) field. Artesian aquifers related to the active faults play an important role in feeding the Aufeis in the Kunlun Mountains. About 120 Aufeis fed on glacier-melt have formed in the West Kunlun Mountains. Icing development was found to vary with the order of river channels and more than half of all of the identified Aufeis are located along first- and second-order river channels. The significance of Aufeis at the QTP related to as an indicator of climate change, and a volume of surface and ground waters conserved into Aufeis should take into consideration of river runoff estimation of the region.

**Keywords:** groundwater icing; Aufeis; permafrost; remote sensing; the Kunlun Mountain; Landsat-8 image

## 1. Introduction

In permafrost regions, one of the main indicators for groundwater occurrence is the development of Aufeis along river valleys [1–3]. In this study, we identified icings as a process and Aufeis (naled) as evidence for groundwater based on one of the most recent review articles [4]. It is more applicable for defining and describing the causal relationship.

Aufeis usually spread in mountainous regions and within the upper parts of river networks [1,4–10]. Icing processes play a significant role in streamflow dynamics. An Aufeis is a redistributor of groundwater flow during the year. Part of the groundwater accumulates as Aufeis in winter, and thus the total winter groundwater flow decreases. However, the melting of Aufeis, especially during springtime, contributes an additional source of water to the surface runoff. It has been estimated that about 4–6% of the total streamflow is contributed by Aufeis ablation in the discontinuous permafrost zones in Canada [11,12]. Aufeis redistribute the flow depending on their modes and supply sources. In addition to supra-permafrost vadose waters and partial intra-permafrost water, seasonal (impermanent, such as seasonal, perennial, or multiyear, icing patches of) icings also exclude a portion of the underflow and sediment runoff, from the hydrological cycle in the winter. According to the study on the water balance for the Firth River basin (Northern

Canada), icings contain up to 30% of the annual groundwater flow, or about 50% of an annual runoff of the river, in the sporadic permafrost zone [2]. In addition to seasonal icings, huge icings also play a role in long-term hydrological and hydrogeological regulations. The layer of groundwater stored in Aufeis can reach 200 mm in precipitation equivalent [13]. Chemical and isotope tracers have been used to separate the sources of the groundwater in the supra- and sub-permafrost and talik waters in the Fish Hole area of the Big Fish River catchment near Aklavik, N.W.T [14].

Regional studies of icings have mostly been conducted in Alaska, Canada, and Russia. A general understanding of the icing distribution in the local watersheds of the Yana, Indigirka, and Kolyma rivers has been obtained [5–7,15]. The main patterns of the icing development in large regions of Russia have also been acquired [8–10,13,16]. Remote sensing techniques enable the calculation of the spatial and temporal dimensions of icy objects. Based on the Cadastre of naleds [15], the change in the areal extent the Aufeis along the Indigirka River watershed has been analyzed [17]. A total of 1213 Aufeis have been detected with a cumulative areal extent to 1287 km$^2$. About 600 new Aufeis have been identified since the 1950s. The important role of winter runoff, which is a function of the antecedent autumn rainfall and the winter warming of air temperatures, are the main factors controlling the icing processes in the Baker Creek research basin of the Northwest Territories [18]. A total of 5500 Aufeis with a cumulative area of about 21,887 km$^2$ have been identified in the Great Slave region around Yellowknife, Northwest Territories [19]. The inter-annual icing dynamics have been attributed to the influences by the meteorological parameters. Numerous regional groundwater springs that form Aufeis have been studied in northeastern Alaska [20].

Only a few studies have been conducted on the Aufeis in China. Based on a numerical simulation of ground thermal regimes, an Aufeis mitigation structure for the cut-slope roadway site in China was designed [21]. It was concluded that an insulating facing wall and a layered insulation drainage ditch are a good choice for preventing the cut slope of the road from forming Aufeis [21]. A study of the Aufeis distribution and its dynamics in the Upper Indus Basin was recently conducted [22]. This river basin is surrounded by the Hindu Kush, Karakoram Mountains and the Greater Himalayan Range. A total of 3700 Aufeis have identified between 4000 and 5500 m a. s. l., and the icing density increases to the east.Studies from the last two decades show environmental changes in mountainous regions due to warming air temperature [23,24] Water supply and irrigation of agriculture relied on snowmelt water within high elevation territories are known [25,26]. Thus, the water management problems of these territories might be solved by the Aufeis as an indicator of groundwater residence, which plays a role in the storage of water. However, icing development should also be vulnerable to climate change because it strongly depends on winter weather settings.

The patterns of icing development and its distribution over the Qinghai-Tibet Plateau (QTP)) are poorly understood. The goal of this study was to create an initial database of the Aufeis on the QTP and to evaluate the patterns of icing processes in the arid and high mountain regions at southern latitudes in the Northern Hemisphere. The spatial distribution, areal extent, and elevational coverage of Aufeis, as well as their dependence on river channel orders, permafrost, and icing dynamics, were investigated using remote sensing images.

## 2. Study Area

### 2.1. Geology of the Kunlun Mountains

The Kunlun Mountains (Figure 1) on the northwestern QTP separate the Tibetan Plateau to the south from the Tarim and Qaidam basins to the north [27]. It extends about 1500 km from west to east and up to 100 km from north to south. The Kunlun Mountains can be tectonically divided into three segments: the West, Middle, and East Kunlun Mountains. These three primary segments can be further sub-divided into northern, middle, and southern sections based on the main tectonic faults [27,28]. The elevation of

the Kunlun Mountains reaches 5000–6500 m with the highest peak a 7167 m a. s. l. Based on the thickness of the molasse deposits in the West Kunlun Mountains at over 5000 m a. s. l., the magnitude of the vertical movement surpasses 10,000–12,000 m [29].

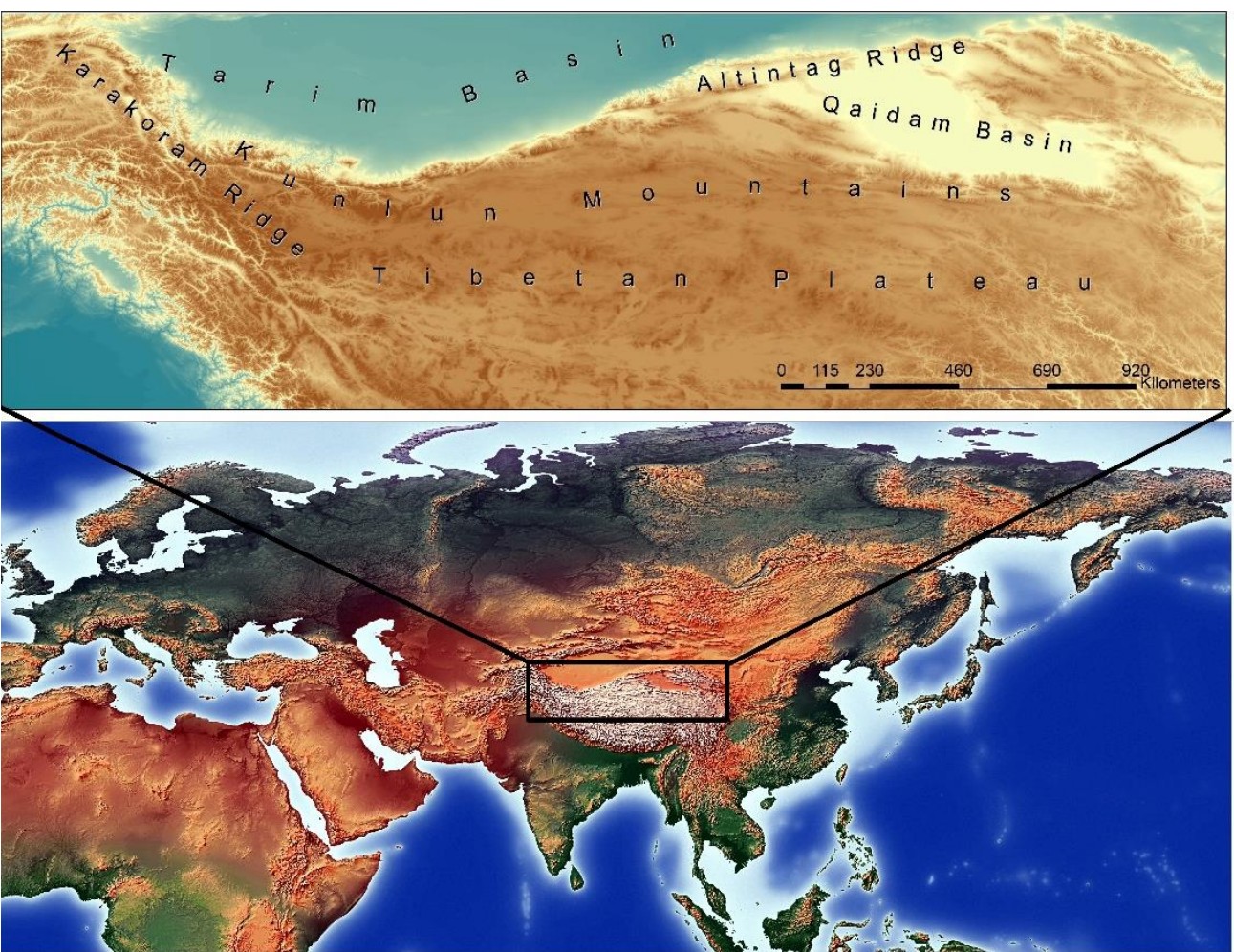

**Figure 1.** Schematic map of the Kunlun Mountains (the data compilation includes data from [30].

The basement rocks of the Kunlun Mountains consist of Precambrian granitic gneisses with Proterozoic granitoids and migmatites. The basement is overlain by Early Paleozoic sedimentary rocks [31]. Owing to horizontal compression, the Kunlun Mountains are exposed to intensive uplifting, and no planation surface processes have developed [27]. Consequently, deep river cutting of the uplifting mountains has occurred. Narrow and deep river valleys with relative elevations of up to 3000–4000 m have formed [27].

Geological structures, such as the Kunlun Mountains, were formed in the zone of tectonic fragmentation as a result of the stretching and down warping of the crust during the Paleozoic. The Kunlun Fault Zone contains large strike-slip faults [32]. These structures underwent collision in the Late Paleozoic, resulting in mountain formation. This type of geological structure is defined as a rift-geosyncline structure [33].

### 2.2. Climate of the Kunlun Mountains

Distinctive atmospheric circulation occurs over the QTP due to its high elevation and topography [34]. Most parts of the Kunlun Mountains have an arid climate, with an annual precipitation of less than 50–200 mm due to its location within the interior of the Eurasian continent and its high elevations [30]. However, some regions, e.g., the western Kunlun Mountains, receive moist-laden air masses from the Arabian sea. This area receives up to 600 mm of precipitation in the form of snowfall or rain above the snowline. The amount

of precipitation decreases from west to east in the Kunlun Mountains and from the high mountains to the deep valleys. On the northern flank of the Kunlun Mountains, the air mass circulation is controlled by the Mongolian-Siberian anticyclone. In summer, these air currents uplift, and the mountainous regions receive up to 300–400 mm of rainfall, while the plain regions receive less than 60 mm of precipitation.

The air temperature in the Kunlun Mountains exhibits altitude zonation [35]. The mean annual air temperature varies from +9.8 °C to −4.9 °C from 2000 to 4500 m a. s. l., and it is 0 °C at approximately 3700 m a. s. l. The minimum and maximum air temperatures reach −35 °C and 16–20 °C at high elevations of >4800 m a. s. l., respectively. The winter air temperature inversion layer occurs at elevations up to 3000 m a. s. l. The daily air temperature range widens with increasing height. Long-term monthly air temperature and precipitation data from 1970 to 2016 for the glacial area of the western Kunlun Mountains were analyzed based on data from meteorological stations [36]. The annual and summer air temperature exhibited increasing trends of 0.24 °C/decade and 0.20 °C/decade from 1970 to 1999, respectively. The annual precipitation was rising at an average rate of about >2 mm/a. It was also found that the snow accumulation increased in the last decade. During the summer, the snowline (equilibrium line altitude, or ELA) is located at 5900–6100 m a. s. l., with an average elevation of about 5930 m a. s. l. [37]. It is controlled by the westerlies, the cold and semi-arid climate conditions, and the precipitation, most of which falls in the summer [38].

*2.3. Hydrogeology of the Kunlun Mountains*

The hydrogeological regime in permafrost regions on the QTP is very sensitive to climate change compared to the Arctic permafrost regions [39]. Glaciers, permafrost, groundwater, lakes, vegetation, wetlands, geologic structures, and tectonic movements are the main factors for influencing the hydrologic and hydrogeologic systems. The hydraulic modes of the groundwater movement may vary significantly from valley to valley, e.g., in the graben basins, owing to the differences in the permafrost parameters (ice content, thickness, and structure). Most of these valleys have favorable conditions for the storage of sub-permafrost groundwater. The active faults disrupt the zonation of the permafrost distribution and are one of the influencing factors for the occurrence of taliks. River and lake taliks are the pathways of groundwater circulation and discharge. Aquifers in the continuous permafrost zone are widespread and extensive, with linear accumulation of groundwater through the river valley, where the fracture zones are distributed [40].

Supra-, intra-, and sub-permafrost groundwaters occur within the permafrost regions in the northern part of the QTP [41]. For example, in the Hei'he River catchment, the supra-permafrost groundwater is mainly fed by local precipitation and glacier meltwater. This water is discharged as baseflow into streams or as springs on the ground surface. Some portion of the supra-permafrost groundwater infiltrates into the sub-permafrost aquifer. Supra-permafrost groundwater has seasonal characteristics, and it is limited in volume during the warm period and until the freeze-up of the active layer. The storage of this type of groundwater is not constant during the warm season due to the laterally uneven thawing of the active layer and the variations in the recharge-discharge modes. Sub-permafrost groundwater is fed by glacier meltwater through lakes and is discharged via the same pathway. Intra-permafrost groundwater is limited to closed taliks. These types of water have a poor hydraulic connection with other types of groundwater. The sub- and intra-permafrost groundwaters mostly occur in the headwater area of the Hei'he River in the Qilian Mountains and is confined to moraines and fluvio-glacial deposits. It has been reported that icings are fed by sub-permafrost artesian groundwater and develops between the narrow gorges and wide plains, where the hydraulic gradient varies significantly [41]. The residence time of the sub-permafrost groundwater may reach 30 years or longer in the area beneath the deeply buried permafrost based on tritium isotope tracer studies conducted on the northern slope of the Kunlun Mountains [39].

More than 300 hydrothermal springs have been identified in the permafrost zone [39]. A couple of boiling spring water overflow within the Kunlun Mountains are known. One of these springs is located near Bouguer Daban peak at 6860 m a. s. l. [42]. The spring water has a temperature of >85 °C and a low salinity (hydrocarbonate sulfate-sodium calcium type).

It has been shown that the groundwater table in the source area of the Yellow River has been lower for the last few decades [43–45]. The source area of the Yellow River is in the discontinuous permafrost zone [45,46]. It has been determined that precipitation, snow-melt water, ice-melt water, and groundwater are the main contributors to the river runoff, and the average discharge of the Yellow River is about 22.6 m$^3$ s$^{-1}$ in the source area [47]. However, the degradation of permafrost due to climate change and human activities is up to 18.6% [34]. This has resulted in the reshaping of the hydrologic environment [48]. When the water level falls below the local river level, some of the river flow is reduced or cut off. For example, in a normal year, the Yellow River along the station section has an annual average runoff of 10 to 20 m$^3$ s$^{-1}$, but during 1991–2004, the annual average runoff decreased to 0.619 m$^3$ s$^{-1}$ (in 2000), which is the lowest on record [39].

### 2.4. Geocryology of the Kunlun Mountains

The area of the QTP underlain by permafrost was estimated to be 126.7 × 104 km$^2$ in the 2000s [49]. The permafrost area of the Tibetan Plateau has reduced in about 115.2 × 10$^4$ km$^2$ by 2015 under climate change and human activity, [50]. The permafrost thickness was calculated to range from 10 m to more than 300 m based on geothermal heat flux [51]. For example, the temperature gradient of permafrost reaches 8 °C/100 m in the Kunlum Mountains. The measured permafrost temperature, or mean annual ground temperature, at the zero annual amplitude depth varies from −3.8 °C to 0 °C. Four types of permafrost zones occur in the Kunlun Mountains according to the Map of the Geocryological Regionalization and Classification in China [52]. Most of areas of the Kunlun Mountains is located in the predominantly continuous permafrost zone. The ridges of the Karakoram, Altyn Tagh, and Qinghai Nanshan mountains are in the alpine permafrost zone. The moderately thick seasonally frozen ground (>1 m) is confined to the lowest elevations of the Kunlun Mountains, i.e., between its the north edge and the Qaidam Basin. The predominantly continuous and island permafrost zone accounts for the smallest proportion of the Kunlun Mountains.

Permafrost has been degrading significantly since the mid-1990s owing to climate change [46,53]. Thus, permafrost in the Kunlun Mountains under a changing climate is more stable compared with that on the southern and eastern QTP. The highest areal extent of permafrost reduction occurred significantly in the warm (>−1 °C) permafrost regions, and lower in the transitory (−2 °C > t < −1 °C) and the cold (<−2 °C) permafrost regions. The areal extent of plateau permafrost decreased by 16% during 1975–2006 with acceleration in 1996–2006 [46].

It should be noted that an important component of the alpine cryosphere in the Kunlun Mountains is glaciers, most of which are concentrated within the western Kunlun Mountains [53]. The glacier area has not significantly shrunk in recent decades [54], and some glaciers have exhibited a positive mass balance [55]. These glaciers are one of the main sources for many inland rivers in the region in summer. It is also known that some icings are recharged by glacier meltwater [41,56–59].

## 3. Materials and Methods

### 3.1. Regional Aufeis Analysis

The Aufeis identification was conducted using the free-access Landsat 8 Operational Land Imager/Thermal Infrared Sensor (OLI/TIRS) Collection 2 Level 2 images. These images were acquired at the time of the maximum icing development. The analysis of air temperature and areal extent of Aufeis indicates that the maximum Aufeis growth occurred

between 10 March and 20 March. A total of 65 images covering the Kunlun Mountains were acquired in March 2018–2020. All chosen images had minimal cloud cover (<20%).

These Aufeis were detected using the normalized difference snow index (NDSI), which was calculated as follows [60].

$$NDSI = (GREEN-SWIR1)/(GREEN + SWIR 1) \tag{1}$$

where GREEN corresponds to band 3 of the Landsat 8 OLI with wavelengths at 0.53–0.59 μm, and SWIR1 corresponds to band 6 of the Landsat 8 OLI with wavelengths at 1.57–1.65 μm. Both bands have a spatial resolution of 30 m.

The image processing was conducted in QGIS 3.16. The areal extent of the Aufeis were determined using the semi-automatic classification technique [17,20]. A threshold value of NDSI about 0.3–0.4 was used for the Aufeis objects based on the atmospheric conditions on the image. Icing development assessment via remote sensing is advantageous for the QTP region, and it has been empirically established. The Aufeis have a distinct contrast with the snow-free surrounding environment at relatively low elevations (below approximately 4000 m according to the satellite images processed in this study). Preliminary analysis has shown that Aufeis develop in the bottom of valleys, in river tributaries, and at the foot of slopes (Figure 2). The Aufeis detection was complicated in the flat landscape of the Qaidam Basin with many lakes. Both objects have similar reflections in snow-free images. The distinction was conducted manually based on several assumptions. First, elongated Aufeis are usually distributed along a stream. Second, Aufeis confined to the foot of slopes and in some cases on valley bottoms have an alluvial cone shape. Third, for gorge sites, the ice-like objects are reflected in the satellite images, and no ice was identified below and above a river (supposed to be open water), so we assumed it was an Aufeis.

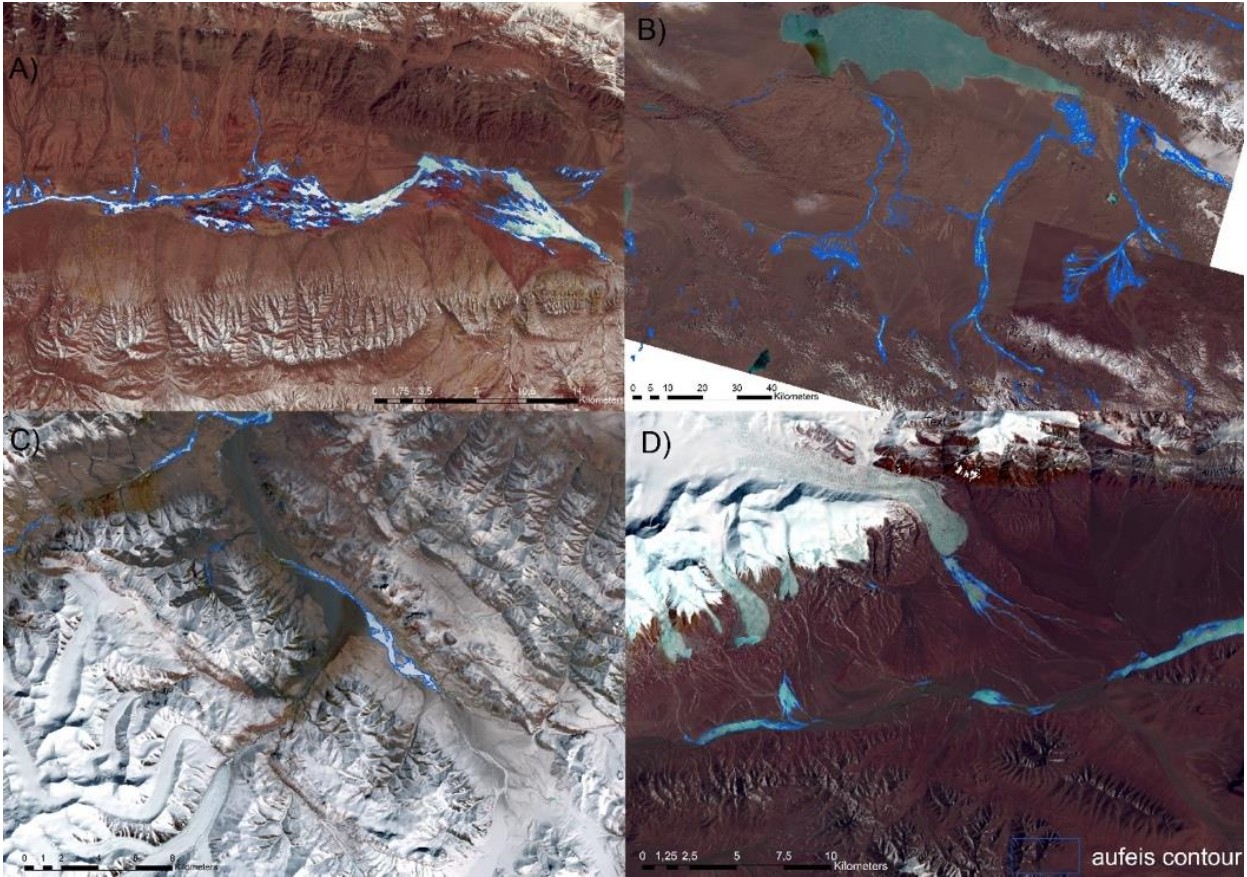

**Figure 2.** Fragments of Landsat 8 OLI imageries (IRGB) with Aufeis contours. Description: (**A**)—typical gigantic Aufeis; (**B**)—groups of sloping Aufeis; (**C,D**)—Aufeis fed by melt glacier water.

The spatial distribution and origin of Aufeis were estimated (Figure 3) using additional data. The Shuttle Radar Topography Mission (SRTM) data were used for the Aufeis georeferencing, which have a spatial resolution of about 30 m along the equator [30]. Each median Aufeis elevation and its areal extent were calculated using QGIS 3.16. The icing belt was statistically evaluated from the largest quantity of Aufeis within each elevation interval (200 m) using Microsoft Excel.

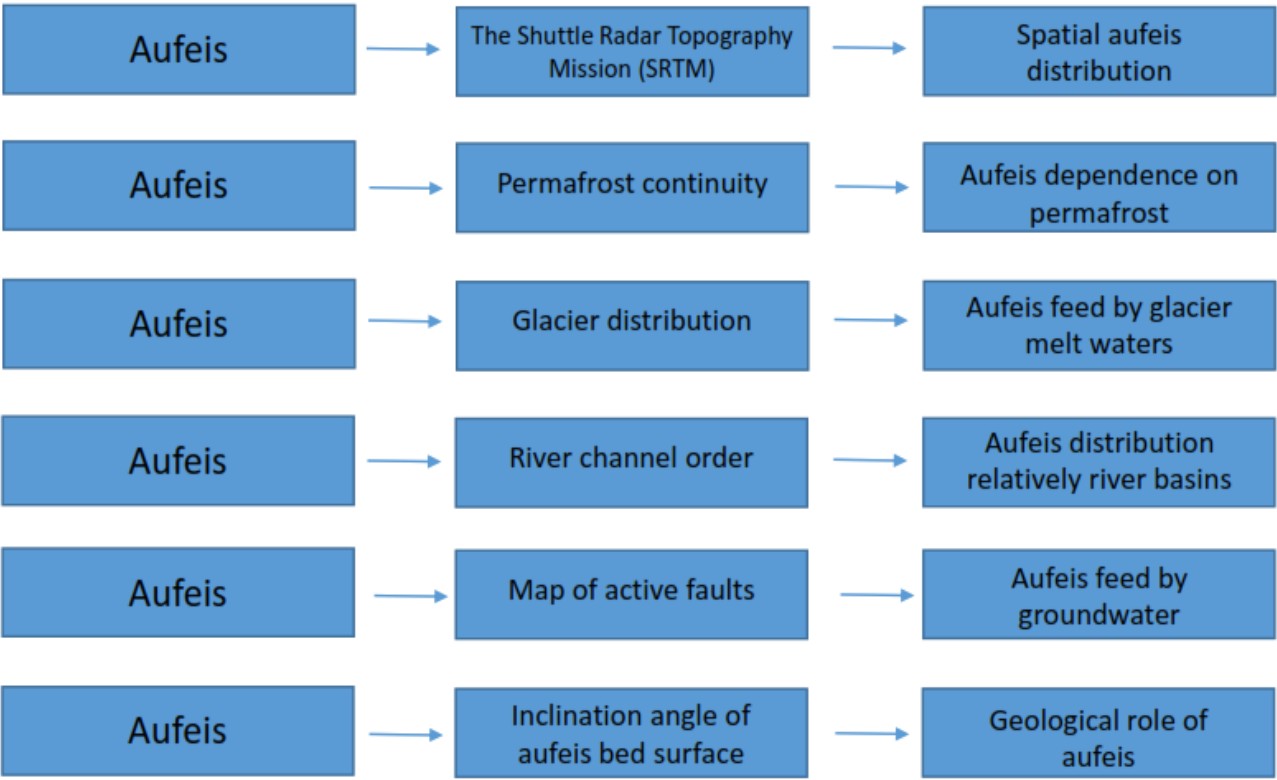

**Figure 3.** Algorithm of Aufeis data analysis.

The slope gradient of river valleys (as an inclination angle of a riverbed) based on SRTM data (Figure 3) using QGIS 3.16 was calculated. Firstly, all of the SRTM imageries (datasets) covering the Kunlun Mountains were merged into the general raster. Secondly, these merged SRTM raster was converted from degree (WGS-84 EPSG: 4326) to the metric (WGS-84/Pseudo-Mercator EPSG: 3857) coordinate reference system. It requires slope gradient calculation, which was done in QGIS 3.16 by the slope tool. Thirdly, the median value of the slope gradient was estimated by zonal statistic plugin applying the Aufeis layer and slope gradient raster.

The Aufeis occurrence was estimated based on the following assumptions: (1) The Aufeis is fed by groundwater if it crosses a fault. The types of water (supra-, intra-, and sub-permafrost waters) were not separated in this study. We did not exclude Aufeis fed by both groundwater and talik water under riverbeds. (2) If the Aufeis is located below a glacier relative to elevation, which recharges the creeks and rivers of the upper part of a river network, then we assumed that it was fed by glacier-melt water. A map of active faults with a scale 1:1,000,000 was used [61]. All glaciers were identified in the Landsat 8 OLI images. The Aufeis locations were conditionally assigned to the three elements of relief: valley bottom, foot of slope, and river tributaries.

The Aufeis distribution was evaluated relative to the areal continuity of permafrost (Figure 3) using the 1:10,000 Map of the Geocryological Regionalization and Classification in China (2021). As was previously mentioned, the Kunlun Mountains contain four geocryological zones: mountain permafrost, predominantly continuous permafrost (70–90%), middle-thick seasonally frozen ground (>1 m), and predominantly continuous and island

permafrost (30–70%). Thereafter, the map was superimposed on the Aufeis layer and the quantity in each geocryological zone was determined using QGIS 3.16.

### 3.2. Applied Classification for Icing Analysis

This study was conducted based on several classifications and assumptions. The areal extents of Aufeis were classified according to the method of Nikolai N. Romanovskii [9] as follows: very small Aufeis ($<10^2$ m$^2$), small Aufeis ($10^2$–$10^3$ m$^2$), medium Aufeis ($10^3$–$10^4$ m$^2$), large Aufeis ($10^4$–$10^5$ m$^2$), very large Aufeis ($10^6$–$10^7$ m$^2$), and gigantic Aufeis ($>10^7$ m$^2$). This differentiation allowed us to preliminarily evaluate the occurrences of Aufeis. As a rule, the very large and gigantic Aufeis were fed by groundwater in the form of springs or both springs and talik.

Strahler's river channel orders (Figure 3) were calculated based on the SRTM using QGIS v.3.16. Eight orders of rivers were extracted in the Kunlun Mountains. However, uncertainties were discovered in the processing due to errors in the calculations in the flat regions, e.g., the Qaidam Basin and southern part of the middle Kunlun Mountains in the flat areas (no slopes), the QGIS's algorithm draws parallel lines for streams. This does not conform to reality. In this study, these regions do not contain many Aufeis. Thus, the Qaidam Basin was excluded in the study, where less than 0.5% of all the Aufeis occur.

## 4. Results

### 4.1. Aufeis Spreadings and Geometry

A total of 1659 Aufeis were identified in the Kunlun Mountains. The altitude distribution of the Aufeis has a pattern. The lowest icing developed in 1900–2000 m a. s. l. (1 Aufeis). This is most likely an exception. The regular elevation for the Aufeis appearance was 2500 m a. s. l. (Figure 4). The upper edge of the Aufeis distribution was 5400 m a. s. l. The icing belt in the Kunlun Mountains is located at elevations of 2500–5400 m a. s. l. Within this belt, a few elevation ranges were distinguished, where the maximum number of Aufeis occurred. The first range is 4300–4600 m a. s. l., which contained about 35% of all the Aufeis in the Kunlun Mountains. The second range (3800–4200 m a. s. l.) is about 29%, and the third (4900–5100 m a. s. l.) is about 9%.

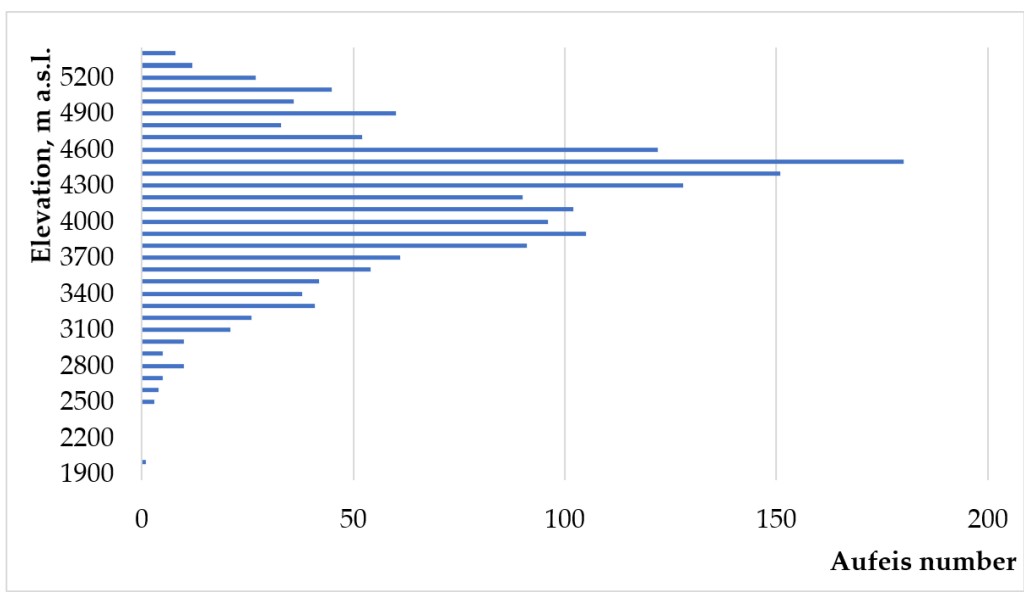

**Figure 4.** Plot of the Aufeis distribution with elevation in the Kunlun Mountains on the northern edge of the Qinghai-Tibet Plateau, West China Mountains.

The total areal extent of the Aufeis in the Kunlun Mountains reaches about 2670 km$^2$. The cumulative areal extent of Aufeis has a different pattern than its distribution relative to the elevation. It is also distributed unevenly. The maximum cumulative area was located at

elevations of 3900–4500 m a. s. l. and covered an area of 1500 km$^2$, accounting for 56% of the total Aufeis area (Figure 5). The second elevation range (4900–5100 m a. s. l.) contained the second largest cumulative Aufeis area of 13%.

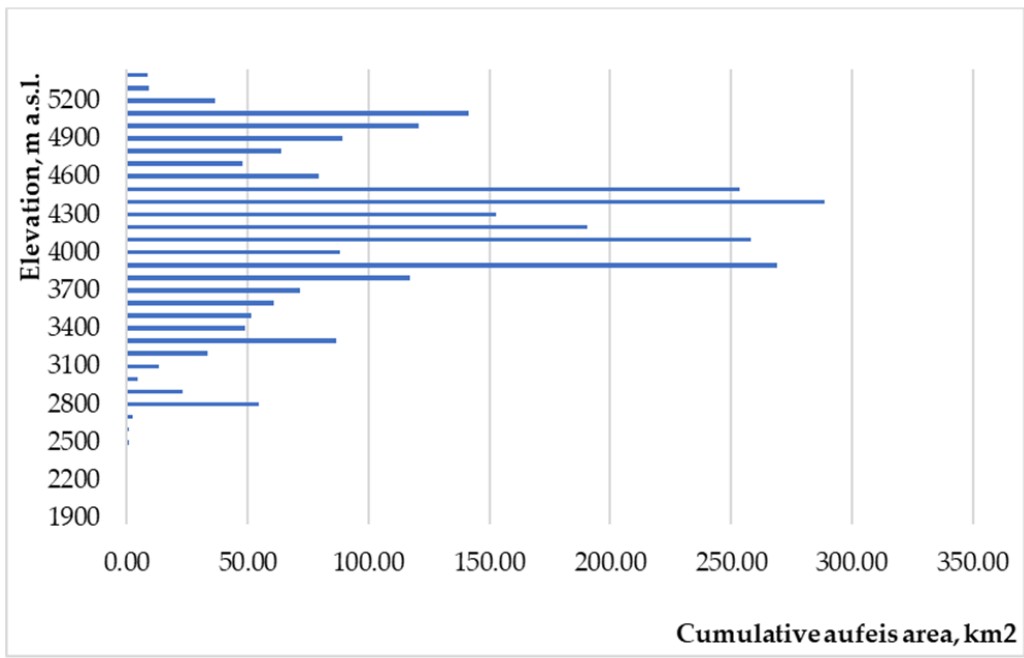

**Figure 5.** Plot of the cumulative Aufeis area distribution relatively to elevation in the Kunlun Mountains on northern edge of the Qinghai-Tibet Plateau, West China.

### 4.2. Icing Relation to Permafrost and Hydrology

A total of 120 Aufeis fed by glacier melt-water were identified. Most of them were located in the western Kunlun Mountains (Figure 2C,D), but some were in the southern and the eastern parts of the eastern Kunlun Mountains. They were mostly concentrated in the upper part of the icing belt. The glacier melt-water fed Aufeis occurred at elevations of 3400–5400 m a. s. l., with the maximum number occurring at 3900–4900 m a. s. l. (Figure 6).

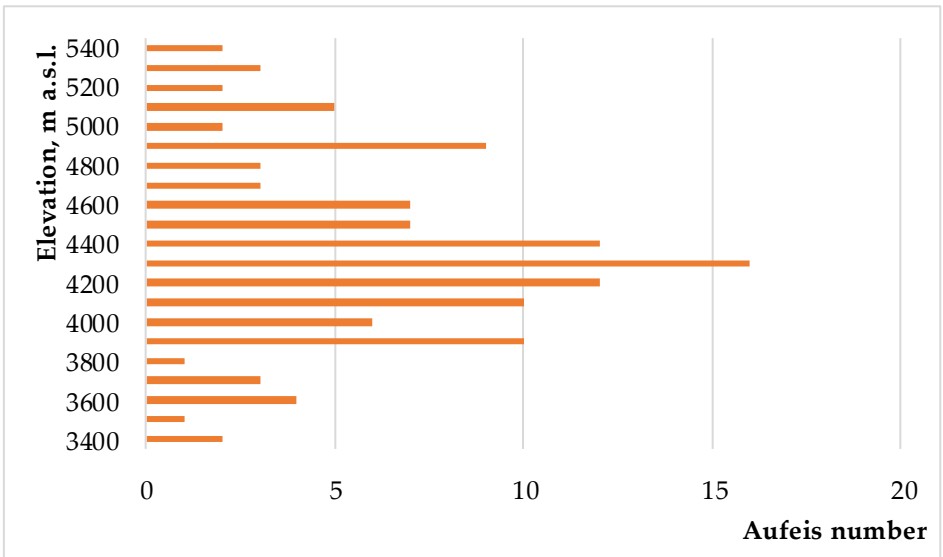

**Figure 6.** Plot of the distribution of the Aufeis fed by glacier melt water relative to elevation in the Kunlun Mountains on the northern edge of the Qinghai-Tibet Plateau, West China Mountains.

The icing distribution relative to the river networks was established (Figure 7). The Aufeis only developed along the upper parts of the first- to seventh-order river channels in the Kunlun Mountains. More than half (~53%) of the Aufeis were located along first- and second-order rivers channels. Only 22% and occurred in river valleys with third- to seventh-order channels. However, the cumulative areal extents of the Aufeis along each river channel order were different. The first-order river channel Aufeis had a cumulative area of 608 km$^2$. The second-order river channel had a cumulative Aufeis area of about 356 km$^2$. In contrast, along the third- to seventh-order river channels, Aufeis had areal extents of 568,587,304, and 182 km$^2$, respectively. Some of the Aufeis were not developed on the bottom of river valleys. Additionally, the slope types of the Aufeis were distinguished. It was found that icing development was the result of the supra- and sub-permafrost discharging onto the ground surfaces on hill slopes [36]. Usually, they develop on alluvial cones or beneath active faults at the slope toes. Most of the sloping Aufeis were concentrated in series with a chain-like shape (Figure 2B), and individual Aufeis were rare. This is one of the reasons for an appearance the large areal extents of the Aufeis determined in this study, that is, because we counted the area of all of the icing surfaces fed on one feeding source (alluvial cone/active fault). The sloping Aufeis accounted for about 25% of the Aufeis, with a cumulative areal extent of about 64 km$^2$.

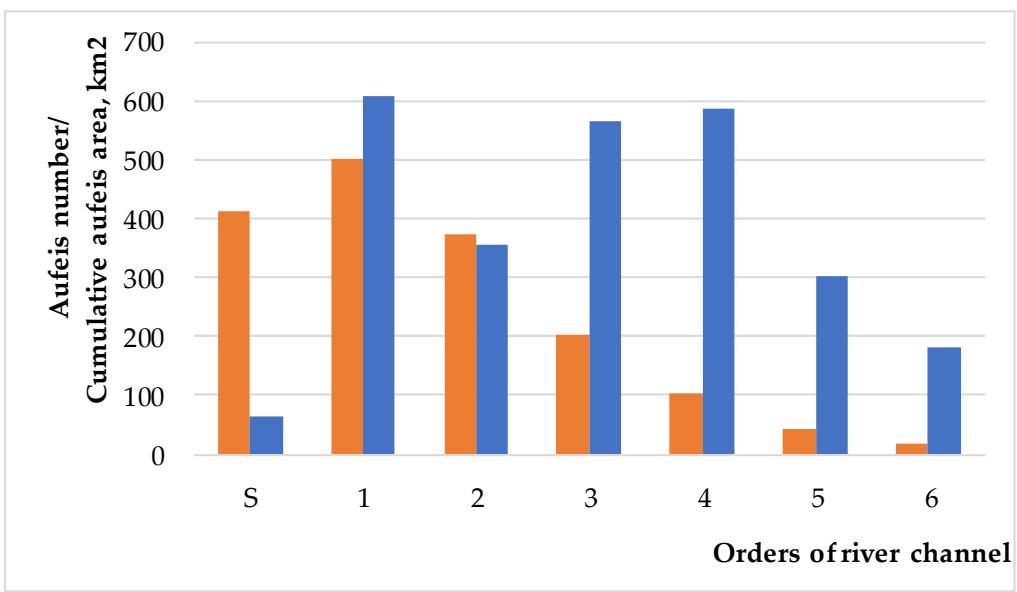

**Figure 7.** Plot of Aufeis number (red) versus cumulative Aufeis area (blue) relative to the orders of the river channels in the Kunlun Mountains. S refers to the sloping Aufeis at the abscissa axis.

The topography of a river valley, especially of a riverbed, plays an important role in the Aufeis appearance. As a result, each countered the Aufeis field's inclination angle was estimated in the current study [61]. Most of the Aufeis (1240) developed on surfaces with incline angles up to 3°. Figure 8 shows the distribution of the Aufeis number with inclination angle for each order of channels. As shown, about 82% of the whole of Aufeis corresponds to the first and second-order channels. These Aufeis develop within all of the ranges of inclination angles. Aufeis development within a higher river valley tend to an almost plain surface up to 1–5 degrees.

The relationship between the Aufeis distribution and permafrost zones was clear in the Kunlun Mountains. Half of the Aufeis area was concentrated in the predominantly continuous permafrost zone. Due to the high elevation and rugged topography of the alpine permafrost zone, the percentage of the frozen-ground covered land area was not defined [62]. The lowest elevation areas of the Kunlun Mountains were in the zone of seasonal frost of more than 1 m in thickness. Slightly more than half (56%) of all the Aufeis

(929) were found in the continuous permafrost zone. The rest of the Aufeis were in the zones of alpine permafrost (33%, 543) and seasonal frost (11%, 186).

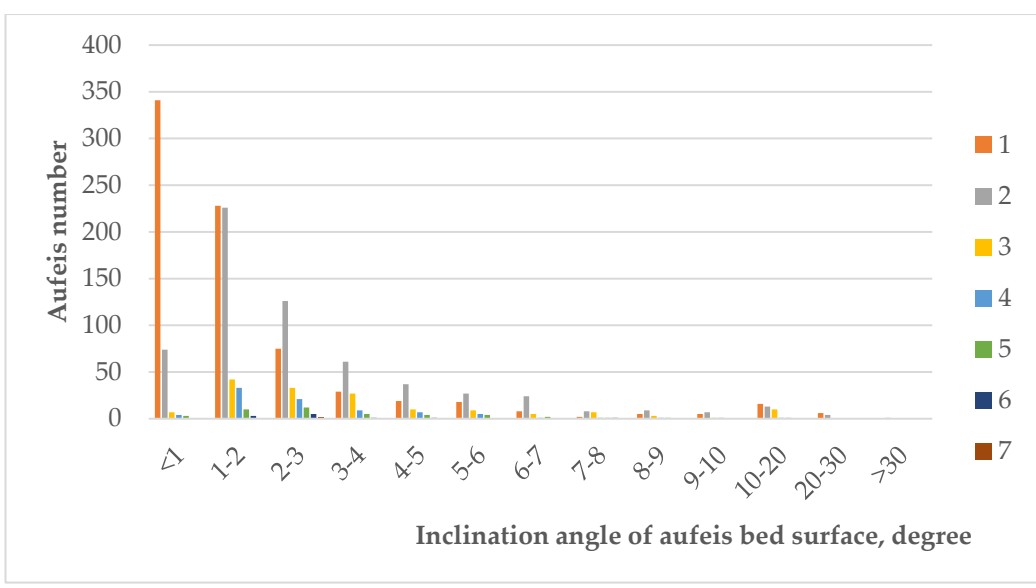

**Figure 8.** Plot of Aufeis number versus inclination angle for Aufeis bed surfaces in the Kunlun Mountains on northern edge of the Qinghai-Tibet Plateau, West China Mountains; 1–7-orders of river channels.

## 5. Discussion

### 5.1. Tectonic and Topography Factors of Aufeis Distribution

The Aufeis were unevenly distributed in the Kunlun Mountains (Figure A1). Most of them were concentrated in the eastern Kunlun Mountains, and fewer were identified in the western Kunlun Mountains. The middle Kunlun Mountains represents a peculiar gap in the icing development. This was caused by the relatively low and rugged topography. Typically, Aufeis are confined to the areas between the mountain regions and intermountain hollows in the Kunlun Mountains, as was also described for the Heihe River Basin [41].

The same pattern of icing development and topography has been detected for many icing regions in Northeastern Russia, Alaska, and Canada [4,8,16,22,62]. Thick fine-grained deposits, low relief, and permafrost occurrence are the main factors for limiting the groundwater springs and consequently for limiting the icing development in the Canadian Shield bedrock [19]. Rugged topography results in high subsurface groundwater velocity, which is favorable for the preservation of water-bearing taliks. In contrast, permafrost settings, freeze-thaw processes of soils, and a sufficient hydraulic water head gradient allow the groundwater to flow out onto the surface and create Aufeis [8]. The same conditions explain the increase in the number of Aufeis at elevations of 2700–3300 m a. s. l. and consequently their higher cumulative areal extent. Most of the icings occurred on plain surfaces surrounded by mountain structures, e.g., the Qaidam and Tarim basins, and fewer occurred in deeply incised river valleys.

The icing belt in the Kunlun Mountains is controlled by a combination of relief, tectonics, permafrost, hydrogeological, and climate conditions. Many of the Aufeis were located at elevations of 4300–4600 and 3800–4200 m a. s. l. (Figure 4). The first range is confined to the middle of the western Kunlun Mountains and the southern flank of the eastern Kunlun Mountains. The second range is in the western part of the West Kunlun Mountains, the northern flank and the eastern part of the eastern Kunlun Mountains, and in the Qinghai Nanshan (South Mountains) Mountains. The Aufeis were concentrated in the third elevation range (4900–5100 m a. s. l.) (Figure 4). The reason for this is the geological structure (complex topography, low air temperature at high elevations, active faults, and apparently, the occurrence of groundwater). The largest numbers of Aufeis

developed on the eastern edge of the West Kunlun Mountains, which is higher than the eastern Kunlun Mountains.

Similar conclusions [22] regarding the altitudinal distribution of Aufeis were reached for the neighbor of the West Kunlun Mountains. This indicates an Aufeis occurrence within the elevation range of 4300–5200 m a. s. l. in the Upper Indus Basin between the Karakoram and Himalayan Mountain ranges. This also demonstrates the significance of cold-arid weather conditions for the icing processes and the altitudinal limitations of their occurrence (4300–5500 m a. s. l.).

Tolstikhin [63] reported that Aufeis mostly developed close to the south-facing slopes at southern latitudes in Siberia and the easternmost regions of the former Union of Soviet Socialist Republics (USSR). Solar radiation is favorable for the occurrence of taliks at very high elevations. The greatest number of Aufeis formed in the headwater areas, tributaries, and mouth regions of the rivers in the mountainous regions in Eastern Siberia [5,7]. The main reason of this icing distribution is the flowing of deeply buried groundwater through the faults [5]. Aufeis are usually located in narrow river valleys, near rapids along river streams, waterfalls, and bedrock. However, the icing locations do not always match the fault position [8]. Frequently, faults predetermine the geomorphology and geological settings of the Aufeis. In this case, the Neotectonics (faults) are an indirect factor for affecting icing processes [8].

Continued tectonic movement on the QTP provides a wide net of active faults, which are the source of the feeding, transition, and discharge of groundwater. In particular, regional sub-permafrost groundwater runoff and its discharge through these faults are favorable for icing development regardless of permafrost continuity [8]. Our analysis indicates that 27% (441) of the Aufeis are confined to the active faults in the Kunlun Mountains. Of these Aufeis, 343 developed on the bottom of the river valley and tributaries and 97 at the slope toes. These 441 Aufeis account for about 39% (1042 km$^2$) of the total areal extent of Aufeis in the Kunlun Mountains. Moreover, 76% of them have very large and gigantic dimensions.

*5.2. Source of Aufeis Recharging*

The gigantic and very large Aufeis usually recharge by sub-permafrost groundwater springs with high debit within continuous permafrost [5–10,63]. These types of ground waters uplift through tectonic faults. Our calculations indirectly reflect the dependence of the Aufeis size on the sub-permafrost groundwater supply and permafrost continuity (Figure 9). However, the number of Aufeis fed by sub-permafrost groundwater through active faults is larger because the icy objects do not necessary cross the lineament directly. The other factors controlling icing development may not be sufficient at a fault zone, and the Aufeis may occur lower along a stream where the conditions are more favorable. Most of the huge Aufeis were confined to the bottoms of river valleys crossed by active faults. As a rule, these places are characterized by sub-permafrost water discharge. It should be noted that we did not exclude Aufeis fed by both sub-permafrost and talik groundwater in that case. Owing to the absence of data about the alluvial deposit thickness and talik continuity, we could not distinguish the water source of the Aufeis. A smaller number of Aufeis developed at the foot of a slope crossed by an active fault. They have large or very large sizes based on the classification scheme used. It should be noted that the areal extent of this type of Aufeis was controlled by the grouping of individual Aufeis confined to general fault/alluvial cone or slopes and concentrated close together.

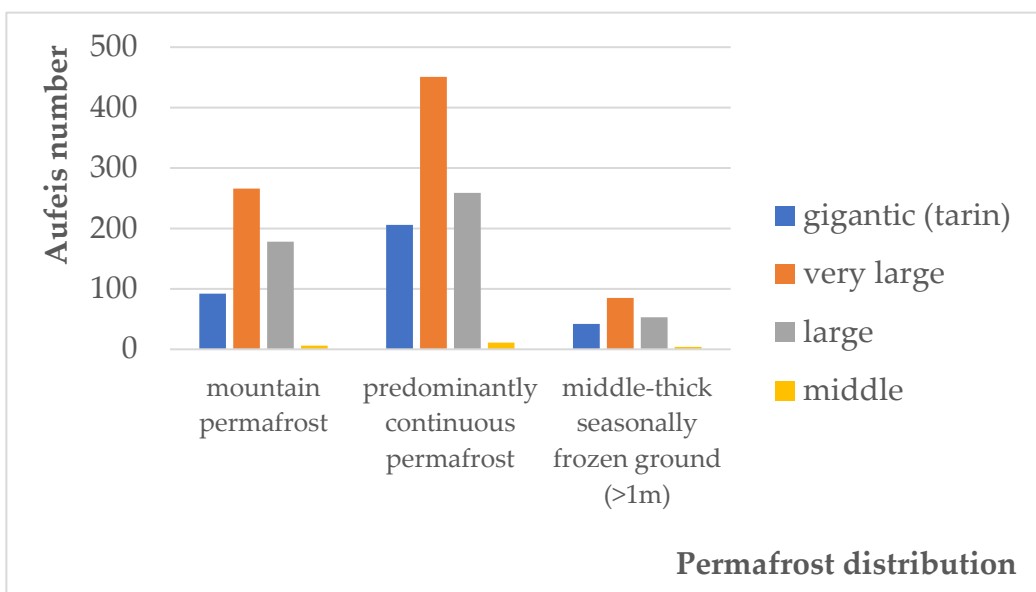

**Figure 9.** Plot of Aufeis number versus types of permafrost distribution in the Kunlun Mountains.

The mechanisms of the groundwater flow contribution to the winter base flow in the upper part of a river network is described here [64,65]. Surface ice and segregation ground ice formation begin unevenly from the headwater to the mouth of a river based on all studies of small watersheds and field observations in the mountainous permafrost regions in the Baikalia, Russia. The river headwater in this region is characterized by highly filtered soils, a rapidly declining groundwater level, and a low groundwater capacity in the pre-winter period. In contrast, typical river mouth areas are characterized by lower groundwater velocities and fine-grained sediments. The decrease in the velocity of the talik groundwater flow is higher than velocity of the freezing boundary in the headwater of a river at the beginning of the winter. However, the decrease in the velocity of the talik groundwater is lower than the velocity of the freezing boundary at the mouth of the river. Thus, the hydrostatic pressure in the sub-river talik aquifer is exceeded, which leads to icing and/or the development of segregation ice. The frozen aquitard grows and its strength gradually increases. When the hydrostatic pressure is exceeded in this case, the aquitard cannot be broken and ice segregation cannot occur. As a result, the area where the hydrostatic pressure is exceeded starts spreading in the upstream direction. The underflow totally freezes and ice formation ceases at the mouth of the river. The frozen-up aquifer gradually moves in the upstream direction and reaches the first-order river channels by the second half of the winter. Finally, in this area, the unfrozen water in the underflow stream starts to overflow onto the surface and icings develop or transform into ice segregation. The mechanism of groundwater overflow has been described earlier by Kane [1].

*5.3. Icing Process Response to Climate Change*

Icing development and its response to climate change should be separated into the western and eastern Kunlun Mountains. The reason for this is that these regions have somewhat different Aufeis feeding regimes. As was previously mentioned, 120 Aufeis are fed by glacier melt-water. The glaciers in the Kunlun Mountains are characterized by low inter-annual ablation. Nevertheless, the ablation of the glaciers during summer feeds the ground water, which in turn recharges the Aufeis. Most of these are located in the western Kunlun Mountains (Figure 2C,D), where many glaciers occur. The Aufeis corresponding to a river channel order have different feeding regimes, i.e., by glacier melt-water and groundwater (Figure 6). Overall, 46% (92) of the Aufeis in the western Kunlun Mountains were fed by glacier melt-water. Almost 95% of these Aufeis are located along first- and second-order rivers and creeks, but a few are located along third-order channels. In contrast, all the Aufeis fed by groundwater are located along fourth- to seventh-order river channels.

The cumulative areal extents of the Aufeis fed by groundwater and glacier melt-water are 469 and 38 km$^2$, respectively. It is known that the groundwater discharge in deeply incised river valleys and icing formations is usually related to regional sub-permafrost flow. Accordingly, they have a long residence time and low vulnerability to climate change. However, glacier melt-water is more sensitive to climate change and may cause Aufeis feeding variations in this region. Additionally, the tendency of shortened of snow cover timing and precipitation increasing, and generally elevation-dependent warming for the Tibetan Plateau/Himalaya region were determined [24]. Thus, rise of groundwater recharging by increasing precipitation should lead to potentially rise of groundwater spring discharge, as a result expansion of the Aufeis sizes in the future. While the cumulative areal extent is not as large (38 km$^2$) relative to the total value, it should transform into a seasonal streamflow redistribution. The last one is an important factor of food-water security of the mountainous region of the Tibetan Plateau, especially due to the decreasing of snowmelt water contribution in a river runoff under high irrigation demand in springtime [26].

*5.4. River Valley Evolution under Icing Process*

Furthermore, we estimated the geological impact on the river valleys. As is shown on Figures 7 and 10, almost all of the ranges of inclination angles correspond to the first and second-order river channels. At that the largest Aufeis develops at the surfaces with an inclination up to 3–4 degrees. These data indirectly indicate that a rough topography is one of the required factors behind the appearance of the Aufeis. As we assume, the headwater region (high Mountains) of a river network (first and second orders) has faster underflow than the lower part (e.g., Aufeis fields), reflecting the angle of the underflow runoff. Therefore, high rates of ground waters are probably a favorable setting for the occurrence of taliks under a river stream. Partial or complete winter freezing of these aquifers facilitates the groundwater overflow and the form of icings. The same pattern was extracted for the inclination angle of the Aufeis surface dependence relative to the areal extent (Figure 10). Almost 87% of the gigantic Aufeis (>1,000,000 m$^2$) were developed on surfaces with inclination angles of up to 2°, and the other 13% had inclination angles of 2–7°. The large and very large Aufeis mostly occurred on slopes of up to 3°, but they cover the entire range of extracted values. Thus, it is reasonable to assume that gigantic Aufeis (tarins) are related to the icing bed morphology in the Kunlun Mountains. The geological structure predetermines the icing distribution [8]. In addition, the plane icing bed surface is a result of its multiyear geological action. Finally, it is confirmed by the fact that the icing field exceeds the annual observable Aufeis area for most of the very large and gigantic Aufeis in the Kunlun Mountains (Figure 2A). The reason for this phenomenon is the migration of the icing processes due to various conditions of ground freezing in winter from year to year and the influences of air temperature on the rate at which the overflow water along a river valley freezes. The geological driver of the Aufeis occurs in the spring when the flood water flows along a river, reaches an icy body, and starts to erode it [16]. Aufeis are mechanically decayed by flowing water, and this decay does not occur in the same place every year due to the variations in it its shape and the water overflow and freezing conditions. This results in furcation of the riverbed and the extension of the riverbanks. This is the reason the stream rate of mountainous rivers declines under multiyear action. Thus, icing fields locally transform the erosion action of a river into a regime characterized by the accumulation of transported sediments in mountainous regions, thereby decreasing the inclination angle of the riverbed.

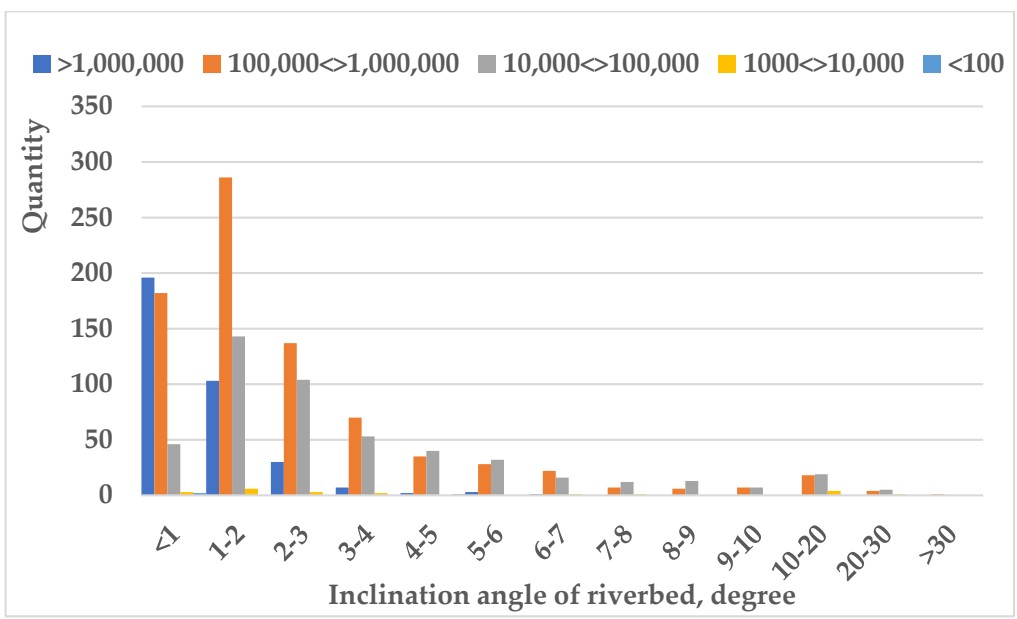

**Figure 10.** Plot of Aufeis number versus inclination angle for Aufeis bed surfaces in the Kunlun Mountains on the northern edge of the Qinghai-Tibet Plateau, West China Mountains Aufeis sizes are reflected by colours (in m$^2$).

The same mechanism of the geological action of icing processes which transformed the erosional activity of rivers was described for the northeastern USSR [7,66,67]. These authors inter-annual and long-term migration of the Aufeis are divided. The first relates to the annual variation of weather parameters, snow thickness and, as a consequence, how the deep talik's aquifer freezes. The long-term Aufeis migration deals with continuous changes of hydrogeological settings, the cryolitic zone and climate change. The dynamics of a hydrogeological and cryolitic zone reflects the variation of the geometry of the talik, its water bearing, water-transmitting capability, and groundwater temperature. Climate change results in conversion of mean annual air temperature, wind speed, and snow accumulation, which determine the degree of talik freezing and intensity of the overflowed water freezing. Both kinds of the Aufeis migration depends on types of permafrost continuity and climate severity. Seasonal or inter-annual migration is featured for discontinuous and sporadic permafrost regions with soft climate settings. These regions are characterized by the wide spread of taliks.Ground waters of these taliks formingthe Aufeis are relatively thinner and shallower. The Aufeis dimension and location controlled mostly by weather conditions are exposed to seasonal variation. In this case its action on a relief is narrow. This is the opposite, for example, to that of eastern Russia where continuous permafrost in mountainous regions means that taliks are confined to tectonic water bearing dislocation. Therefore, icing processes are limited to places with springs marked with these dislocations. Due to the localization of groundwater springs, the groundwater debit is usually higher than within discontinuous or sporadic permafrost zones. The Aufeis dimension increase and its location is fixated. Inter-annual variation of the Aufeis volume is relatively constant due-to the constant annual debit of ground waters filtrated through active faults. The long-term migration of the Aufeis occurs due to climate change or tectonic movements, which influence talik dimension and water capacity. Geological action of the Aufeis in this case is very slow and more significant. A large icy massif in the way of a stream during spring flood acts as a dam. River waters start to erode the Aufeis and bend around it. Thus, lateral erosion intensifies, and banks of rivers degrade more significantly.

## 6. Conclusions

The results of this study reveal several patterns in the icing processes within the Kunlun Mountains. The Aufeis distribution is primarily controlled by the weather conditions,

tectonic structure, topography, hydrogeology, and permafrost distribution. Through the abovementioned factors, it is possible to indirectly link the icing development to altitude zonation. A total of 1659 Aufeis were identified in the icing belt confined to the elevation range of 2500–5400 m a. s. l. Most of the Aufeis developed in three elevation ranges, i.e., 3800–4200, 4300–4600, and 4900–5100 m a. s. l. These concentration zones are controlled by the geological structure and topography of the western and eastern Kunlun Mountains, where the cumulative areal extent of the Aufeis reaches 2670 km$^2$. About 88% of the Aufeis areal extent are gigantic Aufeis, with an area of more than 106 m2. The remaining 11% are large and very large Aufeis, with areas of $10^4$–$10^5$ and $10^5$–$10^6$ m$^2$, respectively. The middle and small Aufeis have areas of $10^4$–$10^3$ m$^2$ and $<10^3$ m$^2$, respectively, accounting for less than 0.2% of the cumulative areal extent.

Active faults play an important role in the feeding and location of the Aufeis. Icing development is related to the orders of the river channels in the Kunlun Mountains. More than half (877) of the Aufeis were located along first- and second-order river channels. The gigantic Aufeis mostly occurred along high order (>3) river channels and on almost planar surfaces with inclination angles of less than 3°. Furthermore, icing has an important geological impact on river valley evolution within the Kunlun Mountains.

The Aufeis fed by groundwater (1539) predominantly occurred within the eastern Kunlun Mountains The remaining 120 Aufeis were fed by glacier melt-water. These Aufeis were confined to the western Kunlun Mountains. All the Aufeis fed by groundwater develop along fourth-order river channels. The Aufeis fed by glacier meltwater are potentially sensitive to climate change.

This study is the first step in understanding the evolution of Aufeis on the QTP at high elevations and at southern latitudes. The icing dynamics in the distinguished altitude intervals, the contribution of climate change to Aufeis development, and the low scale (e.g., longitudinal) patterns should be further and systematically investigated.

**Author Contributions:** Conceptualization, L.G.; Funding acquisition, Q.W.; Investigation, L.G. and Q.W.; Methodology, L.G.; Resources, L.G. and Q.W.; Software, L.G.; Visualization, L.G.; Writing—original draft, L.G.; Writing—review & editing, L.G., Q.W., W.C. and G.J. All authors have read and agreed to the published version of the manuscript.

**Funding:** This research was funded by the Open Fund Project of SKLFSE [Grant No. SKLFSE201701] and CAS President's International Fellowship Initiative (PIFI) for Leonid Gagarin [Grant No. 2019PE0052].

**Data Availability Statement:** Not applicable.

**Acknowledgments:** Authors give acknowledge to Jin Huijun and Ze Zhang for their advice and support during the study.

**Conflicts of Interest:** The authors declare that they have no conflicts of interest. The funders had no role in the design of the study; in the collection, analyses, or interpretation of data; in the writing of the manuscript, or in the decision to publish the results.

**Appendix A**

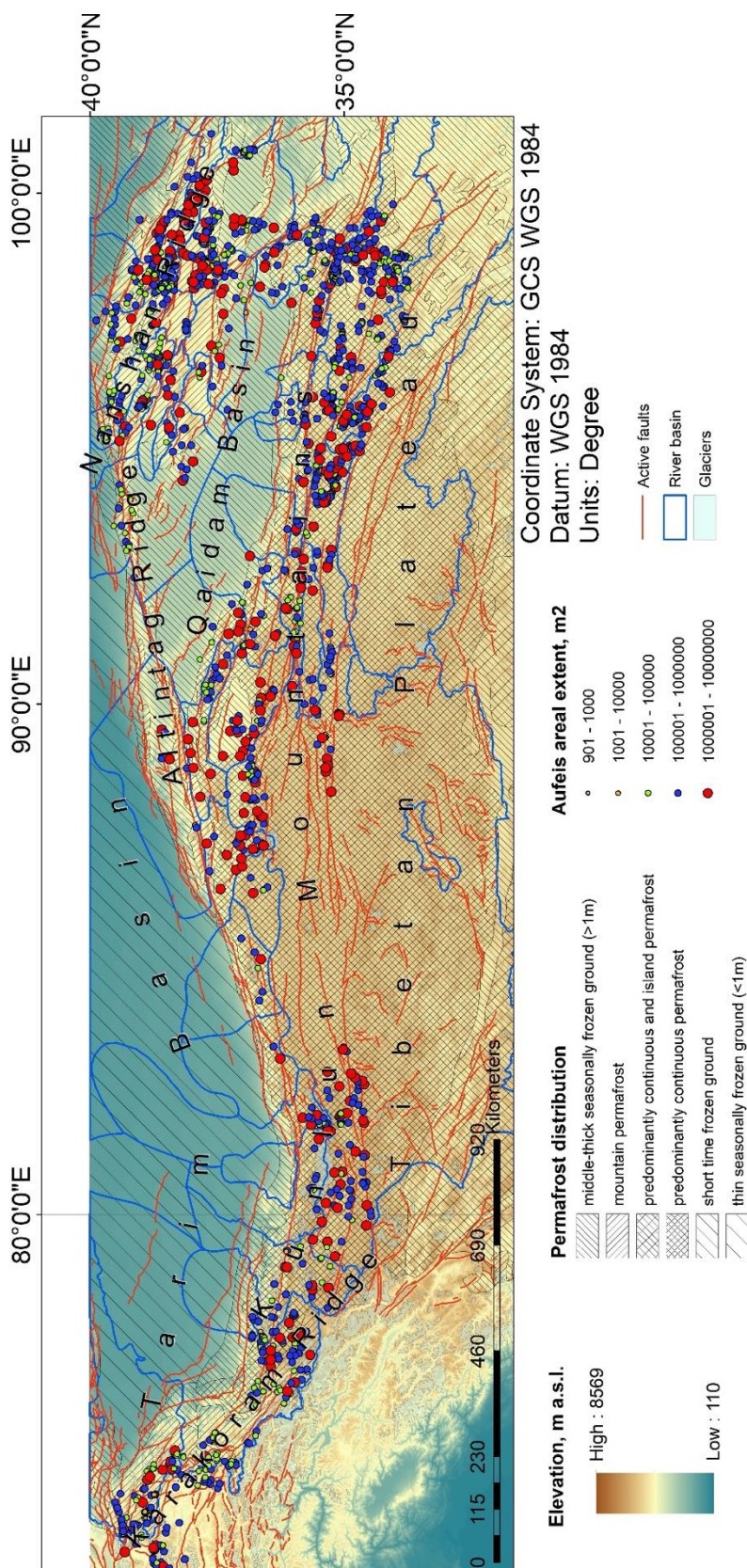

**Figure A1.** Map of Aufeis distribution within the Kunlun Mountains.

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
