# Peer review of "Icings of the Kunlun Mountains on the Northern Margin of the Qinghai-Tibet Plateau, Western China: Origins, Hydrology and Distribution"

_water, doi:10.3390/w14152396_

Round 1

Reviewer 1 Report

The presented article is titled "Icings of the Kunlun Mountains on the northern margin of the Qinghai-Tibet Plateau, Western China: origins, hydrology and distribution". Using Landsat Operational Land Imager images for the period 2017–2020, the authors presented the distribution of the Aufeis and tried to present the factors responsible for their distribution. The problem is most relevant today, in the age of climate change. In addition, noteworthy is the sheer number of Aufeis tested and the enormous area. The paper is interesting and may interest a wide range of scientists, especially interested in processes operating in the periglacial zone. Nevertheless, I have some comments that could add value to the article.

  1. The structure of the text should be improved:
  2. The chapter “Methods” should include the methods by which the slope of the valley was calculated.
  1. Information on slope inclination in the “Discussion” section should first be presented in Results.
  1. In the “Results section”, separate the results from the interpretation.
  2. Discussion should be divided into a few subsections, which, for example, will present, the factors responsible for the distribution of Aufeis in the Kunlun Mountains. In this form, the text is very difficult to read.
  3. In Chapter 5.2 “River valley evolution under icing process”, there are so few examples of changing river valley morphology. I believe that this requires a more detailed discussion of this issue. It seems that the Aufeis change themselves, not the valley.

  1. There is a lack of source data, eg Aufeis size, location and morphology.

In this situation, it isn't easy to verify the presented results.

  1. There is a lack of figures showing exemplary Aufeis. Figure A shows the general distribution of Aufeis. If it is possible to divide them in terms of size, then examples can also be given. There is no geological map with larger faults marked.
  2. What is the relationship of the Yellow River, described precisely with the presented data, with the discussed problem?
  3. Line 426: How is it possible that typical river mouths are characterized by less fine-grained sediments”?.It seems that, on the contrary, the graining of sediments in river estuaries becomes finer.
  4. How should we explain the formation of the described, large >10,000,000 m2, Aufeis? Is it just a matter of simultaneous groundwater outflow or do they take years to develop?

Overall, in my opinion, the manuscript is well written but needs work to be more robust and publishable by WATER MDPI. I recommend that this work is accepted for publication in WATER MDPI but after major revisions are made.

Author Response

Dear Reviewer, 

Thanks a lot for Your attention to our study and important remarks. We have tried take into consideration all of the comments. 

  1. The structure of the manuscript has been improved.
  2. A paragraph about the methods by which the slope of the valley calculation has been added in the chapter “Methods”.
  3. An information on slope inclination in the "Results" has been added.
  4. We have tried to separate the results from the interpretation in the "Results". Some part of the text was moved to the "Discussion".
  5. The "Discussion" chapter have been divided into a few subsections.
  6. In Chapter 5.2 “River valley evolution under icing process” the additional paragraph has been supplemented.

A lack of source data, eg Aufeis size, location and morphology has been filled. 

  1. Figure 2 has been added into the "Method" chapter, and many references in the text have been done to this figure. We think it is very difficult to devide figure A into a few figures. It will result in unclear of perception in the geographic scale. Nevertheless, we are ready discuss it. Actually, we didn't use the geological map in current study. Nevertheless, open source data of the Active Faults of Eurasia Database has employed. We have reflected it on renew Figure A. 
  2. "What is the relationship of the Yellow River, described precisely with the presented data, with the discussed problem?". Of course the Yellow river basin locates toward the East from the Kunlun Mts. But the headwater region of this river lokates on the East boundary the Kunlun Mts., and many Aufeis develop there (we propose to pass the link where it is possible to see it: https://disk.yandex.ru/d/00tF9sP9wq18XA). That is why we used to some data on this river basin.
  3. Line 426: How is it possible that typical river mouths are characterized by less fine-grained sediments”?.It seems that, on the contrary, the graining of sediments in river estuaries becomes finer. You are absolutely right. It is our mistake, we have deleated "less" word.
  4. How should we explain the formation of the described, large >10,000,000 m2, Aufeis? Is it just a matter of simultaneous groundwater outflow or do they take years to develop? We didn't studied the dynamic of icing processes. Possibly, Aufeis might be developed take years, but it is necessary thick layer of ice, (e.g. the same conditions are characterized to southern Yakutia and Far East of Russia) which will melt during whole summer time. But in the setting of low latitude and high elevation where solar activity is more stronger, it is unlikely. In a different from southern Yakutia or Far East of Russia, summer time duration is wider at the QTP. We wrote in the manuscript, some of chain of Aufeis we have combined to the whole icy body, because it was difficult distinguish them as separate aufeis. For example, along a river valley two or more icy body develops in a distance one another about a few hundreds meters. We was combined them to single Aufeis. We think that similar Aufeis recharges by groundwater.

Reviewer 2 Report

The manuscript addresses the formation and spatial distribution of Aufeis on the Qinghai-Tibet Plateau using remote-sensing datasets. This manuscript does not like an original research and is more like a chapter book reviewing the geology, climatology, and hydrology settings of the Kunlun Mountains. Additionally, the "Methodology" and “Results and discussions” sections lack novelty.

Author Response

Dear Reviewer, 

Thanks a lot for Your attention to our study and remarks. We take note Your comment. The structure was chosen due-to more clearly viewing for readers on environmental settings of the Kunlun Mts and its influence on icing development as we think. We agree that methodic of the study has not some novelty. Opposite, we used to use similar methodology as our colleagues, who study the Aufeis, that to have opportunity to compare our results and generalize them in the future. "Results" and "Discussion" chapters have novelty we think. Aufeis distribution and its spatial analizing for the Kunlun Mts was done first. Nobody did it before. The last decade tendency of the Aufeis study is drafting of the Aufeis catalog for different regions, which allow understand icing processes pattern in geographic scale.

Round 2

Reviewer 2 Report

I still believe that the novelty of the work has not been justified. Nonetheless, it may be interesting for some local readers. Authors can at least improve the work by addressing the importance of mountainous areas for food-water security and the climate change impacts in these environments on the globe. I suggest citing the followings:

Nouri M, Homaee M (2021) Spatiotemporal changes of snow metrics in mountainous data-scarce areas using reanalyses. J Hydrol. 10.1016/j.jhydrol.2021.126858

Qin Y, Abatzoglou JT, Siebert S, Huning LS, AghaKouchak A, Mankin JS, Hong C, Tong D, Davis SJ, Mueller ND (2020) Agricultural risks from changing snowmelt. Nat Clim Change 10:459-465. 10.1038/s41558-020-0746-8

Rhoades AM, Jones AD, Ullrich PA (2018) Assessing Mountains as Natural Reservoirs With a Multimetric Framework. Earth's Future 6:1221-1241. 10.1002/2017ef000789

Hatchett BJ, Rhoades AM, McEvoy DJ (2021) Monitoring the Daily Evolution and Extent of Snow Drought. Nat. Hazards Earth Syst. Sci. Discuss. 2021:1-29. 10.5194/nhess-2021-193

Serquet G, Marty C, Dulex J-P, Rebetez M (2011) Seasonal trends and temperature dependence of the snowfall/precipitation-day ratio in Switzerland. Geophys Res Lett 38:n/a-n/a. 10.1029/2011gl046976

Mountain Research Initiative E. D. W. Working Group (2015) Elevation-dependent warming in mountain regions of the world. Nat Clim Change 5:424-430. 10.1038/nclimate2563

Hock R, Rasul G, Adler C, Cáceres B, Gruber S, Hirabayashi Y, Jackson M, Kääb A, Kang S, Kutuzov S, Milner A, Molau U, Morin S, Orlove B, Steltzer H (2019) High Mountain Areas. in Pörtner H-O, Roberts DC, Masson-Delmotte V, Zhai P, Tignor M, Poloczanska E, Mintenbeck K, Alegría A, Nicolai M, Okem A, Petzold J, Rama B, Weyer NM (eds.) IPCC Special Report on the Ocean and Cryosphere in a Changing Climate. Working Group II Technical Support Unit, governmental Panel on Climate Change.

Author Response

Dear Reviwer, Thanks a lot for Your attention to our study. 

We have a look at suggested papers and have agreed with improvement the work by addressing the importance of mountainous areas for food-water security and the climate change impacts. The paragraph about climate change influence on water management problems in mountainous regions in introduction and few sentences in sub-heading 5.3 of discussion were added. 

This manuscript is a resubmission of an earlier submission. The following is a list of the peer review reports and author responses from that submission.